

# Spatial patterns and sociodemographic predictors of chronic obstructive pulmonary disease in Florida

Sara Howard and Agricola Odoi

Biomedical and Diagnostic Sciences, University of Tennessee, Knoxville, TN, United States of America

## ABSTRACT

**Background**. Chronic obstructive pulmonary disease (COPD) is a chronic, inflammatory respiratory disease that obstructs airflow and decreases lung function and is a leading cause death globally. In the United States (US), the prevalence among adults is 6.2%, but increases with age to 12.8% among those 65 years or older. Florida has one of the largest populations of older adults in the US, accounting for 4.5 million adults 65 years or older. This makes Florida an ideal geographic location for investigating COPD as disease prevalence increases with age. Understanding the geographic disparities in COPD and potential associations between its disparities and environmental factors as well as population characteristics is useful in guiding intervention strategies. Thus, the objectives of this study are to investigate county-level geographic disparities of COPD prevalence in Florida and identify county-level socio-demographic predictors of COPD prevalence.

**Methods**. This ecological study was performed in Florida using data obtained from the US Census Bureau, Florida Health CHARTS, and County Health Rankings and Roadmaps. County-level COPD prevalence for 2019 was age-standardized using the direct method and 2020 US population as the standard population. High-prevalence spatial clusters of COPD were identified using Tango's flexible spatial scan statistics. Predictors of county-level COPD prevalence were investigated using multivariable ordinary least squares model built using backwards elimination approach. Multicollinearity of regression coefficients was assessed using variance inflation factor. Shapiro-Wilks, Breusch Pagan, and robust Lagrange Multiplier tests were used to assess for normality, homoskedasticity, and spatial autocorrelation of model residuals, respectively.

**Results**. County-level age-adjusted COPD prevalence ranged from 4.7% (Miami-Dade) to 16.9% (Baker and Bradford) with a median prevalence of 9.6%. A total of 6 high-prevalence clusters with prevalence ratios >1.2 were identified. The primary cluster, which was also the largest geographic cluster that included 13 counties, stretched from Nassau County in north-central Florida to Charlotte County in south-central Florida. However, cluster 2 had the highest prevalence ratio (1.68) and included 10 counties in north-central Florida. Together, the primary cluster and cluster 2 covered most of the counties in north-central Florida. Significant predictors of county-level COPD prevalence were county-level percentage of residents with asthma and the percentage of current smokers.

**Conclusions**. There is evidence of spatial clusters of COPD prevalence in Florida. These patterns are explained, in part, by differences in distribution of some health behaviors (smoking) and co-morbidities (asthma). This information is important for guiding

Corresponding author
Agricola Odoi, aodoi@utk.edu

intervention efforts to address the condition, reduce health disparities, and improve population health.

## INTRODUCTION

Chronic obstructive pulmonary disease (COPD) is a chronic respiratory condition in which inflammation obstructs airflow and decreases lung function (*Easter et al., 2020*). Globally, COPD was ranked as the third leading cause of death in 2019 (*World Health Organization, 2023*) and had a prevalence of 2,638.2 cases per 100,000 people in 2019 (*Safiri et al., 2022*). In the United States (US), the prevalence among adults is 6.2%, but increases with age to 12.8% among those 65 years or older (*Wheaton et al., 2019*). The condition is ranked among the top 10 causes of death in the US (*Syamlal, 2022*). Economically, the financial costs of COPD are steadily increasing. From 2002 to 2010, the annual direct cost of COPD in the US increased from $18 billion to $29.5 billion (*Hillas et al., 2015*), and more recent calculations estimate the annual direct cost at $32 billion (*Guarascio et al., 2013*).

Several environmental and sociodemographic factors have been shown to be associated with COPD including air pollutants (*Huang et al., 2019*), rurality (*Croft et al., 2018*; *Raju et al., 2019*), and poverty (*Raju et al., 2019*). Chronic exposure to outdoor air pollutants including particulate matter 2.5 ($PM_{2.5}$) have been shown to reduce lung function and increase COPD risk (*Huang et al., 2019*). Interestingly, COPD prevalence is reportedly higher in rural areas (8.2%) compared to metropolitan areas (4.7%) (*Croft et al., 2018*). This is surprising because rural areas tend to have better air quality with the mean total number of $PM_{2.5}$ days greater than the US Environmental Protection Agency's National Ambient Air Quality Standard at 0.95 days compared to 11.21 days in metropolitan areas (*Strosnider et al., 2019*). However, rural communities often suffer from poor indoor air quality, including exposure to second-hand smoke and solid fuels used as heating sources. Populations in rural areas may also participate in occupations such as mining that increase risk of harmful respiratory exposures that may contribute to higher COPD prevalence (*Croft et al., 2018*; *Raju et al., 2019*; *Raju et al., 2020*). Other sociodemographic characteristics such as poverty have been reported to be associated with higher COPD prevalence including both community-level poverty (OR: 1.12, 95% CI [1.03–1.32]) and household poverty (OR: 1.08, 95% CI [1.06–1.10]) (*Raju et al., 2019*).

In the US, Florida has one of the largest populations of older adult residents, accounting for 4.5 million adults 65 years or older (*United States Department of Health and Human Services A for CL, 2021*), which makes Florida an ideal geographic location for investigating COPD since disease prevalence increases with age (*Wheaton et al., 2019*). In 2021, COPD resulted in 77,506 emergency department visits and 25,405 hospitalizations in Florida (*Florida Department of Health, 2022*). These frequencies equate to an age-adjusted prevalence of 32.8 COPD related emergency room visits per 10,000 persons, and 10.2 COPD

related hospitalizations per 10,000 persons (*Florida Department of Health, 2022*). However, even with an older state population and the reported burden of COPD in Florida, there is limited information on the association between COPD and environmental factors and population characteristics in Florida. No studies have examined the association between COPD and area-level characteristics such as environmental and sociodemographic factors in the state. These types of area-level studies are useful for not only identifying predictors of COPD risk but also in identifying possible geographic disparities. The information gleaned from such studies can be helpful in the creation of targeted public health prevention and control efforts. Thus, the objectives of this study were to investigate county-level spatial patterns of COPD prevalence in Florida and identify county-level sociodemographic predictors of the identified spatial patterns of COPD prevalence.

## MATERIALS & METHODS

### Ethical statement

This study was reviewed by the University of Tennessee, Knoxville Institutional Review Board which determined that it was not human subjects' research (IRB Number: UTK IRB-23-07928-XM). Therefore, it determined that IRB oversight was not required.

### Study area

This ecological study was performed in Florida, which has an estimated population of 21.2 million people (*US Census Bureau, 2020a*; *US Census Bureau, 2020b*) spread across 67 counties, 44.7% of which are considered rural (*Florida Department of Health P and PR, 2016*). The most populated county is Miami-Dade with 2,830,500 residents, and the least populated is Lafayette County with 8,613 residents (*Florida Department of Health, 2019*). Approximately 80.1% of the Florida residents are at least 18 years old (*US Census Bureau, 2020a*; *US Census Bureau, 2020b*). Of these, 41.6% are 18–44 years old, 32.8% are 45–64 years old, and 25.6% are ≥65 years old (*US Census Bureau, 2020a*; *US Census Bureau, 2020b*). Florida has a slightly lower percentage of males (48.9%) compared to females (51.1%) (*Florida Department of Health, 2019*). Individuals of White race account for 76.9% of the population whereas those who are Black account for 17.0% and other races account for 6.0% (*US Census Bureau, 2020a*; *US Census Bureau, 2020b*). Residents of Hispanic or Latino ethnicity comprise 26.8% of the population (*US Census Bureau, 2020a*; *US Census Bureau, 2020b*).

### Data sources

The 2019 county-level crude prevalence of COPD for each age group (18–44 years old, 45–64 years old, and ≥65 years old) were obtained from Florida Health CHARTS (*Florida Department of Health, 2019*). Sociodemographic variables investigated for possible association with county-level COPD prevalence were also downloaded from Florida Health CHARTS (*Florida Department of Health, 2019*) and County Health Rankings and Roadmaps (Table 1) (*County Health Rankings & Roadmaps, 2019*). Cartographic boundary files, for map generation, were obtained from the United States Census Bureau TIGER Geodatabase (*United States Census Bureau, 2020*).
**Table 1  Potential predictors of chronic obstructive pulmonary disease prevalence.**

| Variable theme | Variable |
| --- | --- |
| Health | % Asthma Diagnosis |
| Smoking | % Current Smokers |
| | % Former Smokers |
| | % Never Smokers |
| Economic | % Individuals Below Poverty Line |
| | % Families Below Poverty Line |
| | Median Household Income |
| Sex | % Male |
| | % Female |
| Race and Ethnicity | % White Hispanic |
| | % White Non-Hispanic |
| | % Non-White Hispanic |
| | % Non-White Non-Hispanic |
| Geographic | % Rural |
| | Mean Daily Particulate Matter 2.5 ($PM_{2.5}$) |

## Descriptive analyses and data preparation

All descriptive analyses and data preparation were performed using RStudio 2021.09.2 (*RStudio Team, 2020*) interface of R version 4.0.4 (*R Core Team, 2020*). County-level COPD prevalence was age-standardized using the direct method (*Dohoo, Martin & Stryhn, 2012*) and 2020 United States population as the standard population (*US Census Bureau, 2020a*; *US Census Bureau, 2020b*). The normality distribution of continuous variables was assessed using the Shapiro–Wilks test (*Royston, 1983*). Since the majority of the continuous variables were not normally distributed, medians and interquartile ranges were reported for all continuous variables.

## Cluster identification and investigation

Tango's flexible spatial scan statistic (FSSS) was used to detect and identify high-prevalence spatial clusters of COPD using FleXScan (*Tango & Takahashi, 2005*). Specification of FSSS included maximum scanning window of 15 counties and Poisson probability models with a restricted log likelihood ratio (LLR) tests and an alpha of 0.2. Significance testing was performed using 999 Monte Carlo permutations and a critical *p*-value of 0.05. Only statistically significant clusters (*p*-value < 0.05) with a relative risk >1.2 were reported to avoid reporting clusters with very low relative risks. The choice of 1.2 cut-off was based on similar interpretations from previous published studies (*Prates, Kulldorff & Assunção, 2014*; *Lord, Roberson & Odoi, 2021*; *Khan, Odoi & Odoi, 2023*).

## Assessment of correlations among potential predictors

Spearman's rank correlation coefficient was used to identify highly correlated ($|r| \geq 0.7$) potential predictors of COPD prevalence. Only one of a pair of highly correlated variables was retained for assessment in the subsequent multivariable model. The decision of which

of a pair of highly correlated variables to retain was based on biological and statistical considerations.

## Predictors investigation of geographic distribution of COPD prevalence

Ordinary least squares (OLS) regression was used to identify county-level predictors of age-adjusted COPD prevalence using RStudio 2021.09.2 (*RStudio Team, 2020*) interface of R version 4.0.4 (*R Core Team, 2020*). A conceptual model was developed and used to guide model development and selection of variables to include in the investigation (Fig. 1). The multivariable OLS model was built in two steps. In the first step, univariable models identified potential predictors that had univariable associations with age-adjusted COPD prevalence at a relaxed critical *p*-value of 0.2. Only potential predictors with a *p*-value <0.2 were considered for investigation in the multivariable model. In the second step, a multivariable OLS model was built using manual backwards selection and a critical *p*-value of 0.05. A variable was considered a confounder if its removal from the OLS model resulted in a >20% change in the OLS regression coefficient of at least one of the other variables in the model. The above approach of identification was used in conjunction with the conceptual diagram shown in Fig. 1. Multicollinearity was assessed using variance inflation factor (VIF) with values >10 considered indicative of multicollinearity. The Shapiro–Wilks test (*Royston, 1983*) and the Breusch-Pagan test (*Breusch & Pagan, 1979*) were used to assess normality of residuals and heteroskedasticity, respectively (*Dohoo, Martin & Stryhn, 2012*). The VIF, Shapiro–Wilks test, and Breusch-Pagan test were all implemented in R version 4.0.4 (*R Core Team, 2020*) with the RStudio version 2021.09.2 (*RStudio Team, 2020*) interface. The Robust Lagrange Multiplier test, implemented in GeoDa (*Anselin, Syabri & Kho, 2006*), using queen weights was used to assess spatial autocorrelation of the ordinary least squares regression model residuals.

## Cartographic displays

All cartographic displays were generated in ArcGIS (Esri, Redlands, CA, USA). The spatial distribution of county-level age-adjusted COPD prevalence as well as its significant county-level predictors were displayed in choropleth maps. The critical intervals of the choropleth maps were determined using Jenk's optimization classification scheme (*Brewer & Pickle, 2002*). Spatial distribution of the identified significant high-prevalence clusters of COPD were also displayed in a map.

## RESULTS

### Geographic distribution of COPD prevalence

The county-level age-adjusted COPD prevalence ranged from 4.7% (Miami-Dade) to 16.9% (Baker and Bradford) (Figs. 2 and 3) with a median prevalence of 9.6%. The highest prevalence proportions were observed in seven counties, six of which Baker, Bradford, Citrus, Dixie, Lafayette, and Suwannee were located near north-central Florida. The remaining county, Walton, was in the panhandle. The lowest prevalence was observed among seven counties, four of which Collier, Miami-Dade, Pasco, and Sarasota were

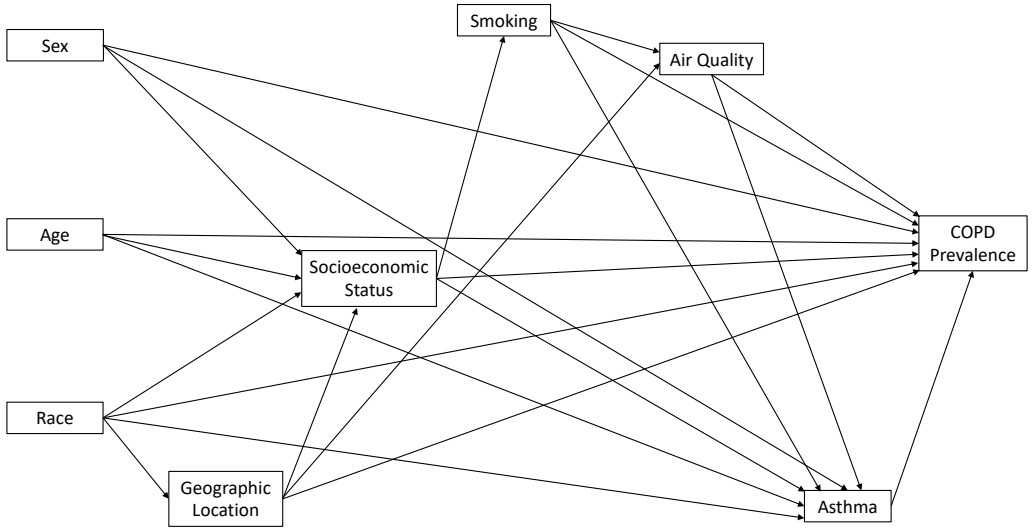

**Figure 1** Conceptual model representing predictors of chronic obstructive pulmonary disease in Florida.

located in the south and south-central portions of the state. Two counties (Alachua and Seminole) were in the north-central Florida, and the other county (Wakulla) was located in the panhandle area (Figs. 2 and 3).

## High-prevalence clusters of COPD

A total of six high-prevalence clusters with prevalence ratios >1.2 were identified (Table 2 and Fig. 4). The primary cluster, which was also the largest geographic cluster with 13 counties, stretched from Nassau County in north-central Florida to Charlotte County in south-central Florida (Figs. 2 and 4). However, cluster 2 had the highest prevalence ratio (1.68) and included 10 counties in north-central Florida. Together, the primary cluster and cluster 2 covered most of the counties in north-central Florida (Figs. 2 and 4). Additionally, two other clusters (clusters 3 & 4) covering large geographic areas were identified. Cluster 3 covered seven counties in south and south-central Florida and had a prevalence ratio of 1.24 whereas cluster 4 had a prevalence ratio of 1.54 and included seven counties in the panhandle area.

## Predictors of COPD prevalence

All investigated potential predictors, except percentage of former smokers and mean daily $PM_{2.5}$, had univariable associations ($p < 0.2$) with county-level COPD prevalence (Table 3) and were assessed in the multivariable model. Based on the results of the final multivariable model, only percentage of residents with asthma and percentage of current smokers were significant predictors of county-level COPD prevalence (Table 4). The percentage of non-White non-Hispanic residents was included in the final model as a confounder for the association between percentage of residents with asthma and COPD prevalence. The final multivariable OLS model accounted for 62.1% (adjusted $R^2 = 0.621$) of the variation in

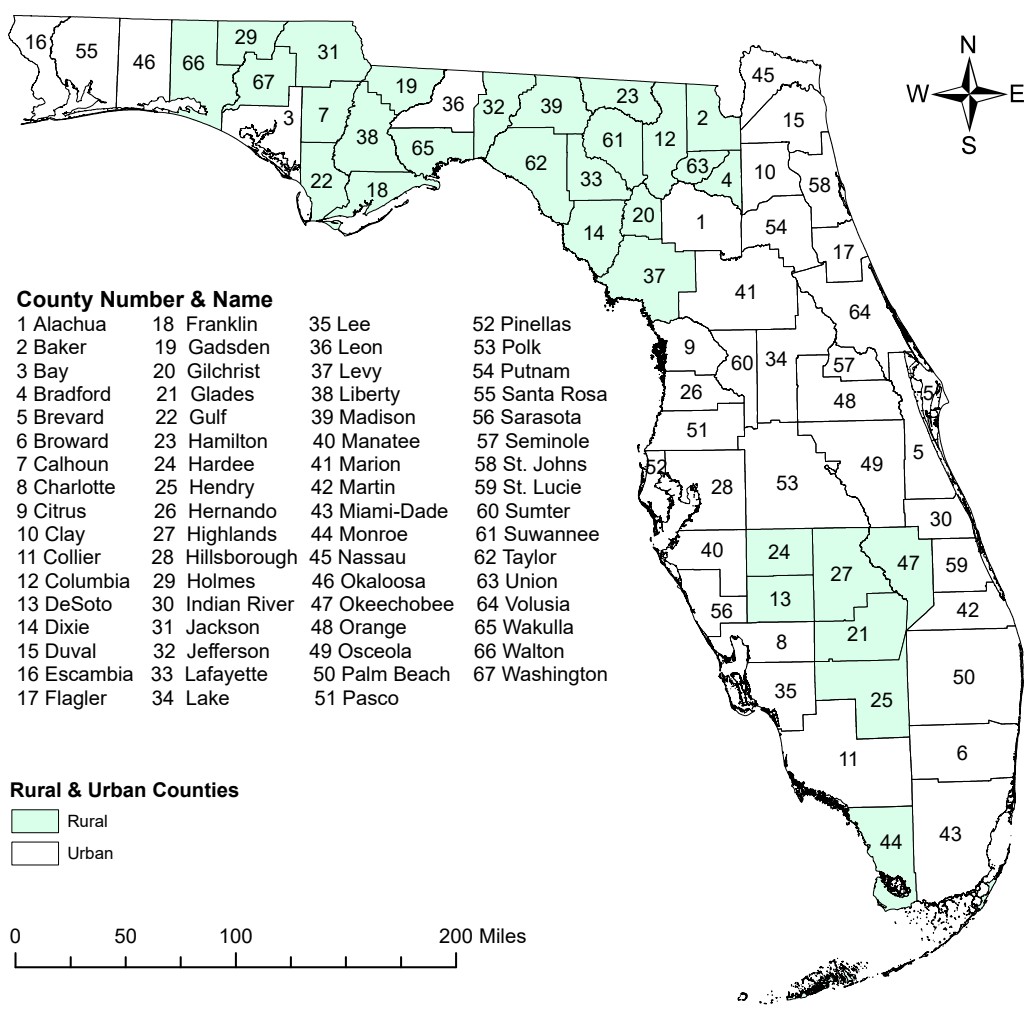

**Figure 2 Geographic distribution of rural and urban counties in Florida.** The base maps were downloaded from the United States Census Bureau: https://www.census.gov/geographies/mapping-files/time-series/geo/tiger-line-file.2019.html; License: Public Domain.

COPD prevalence and showed no evidence of non-normally of residuals (Shapiro–Wilks test $p = 0.805$), multicollinearity (all VIFs < 5), or heteroscedasticity of residuals (Breusch Pagan test $p = 0.191$). There was also no evidence of spatial dependence of residuals based on both the robust Lagrange Multiplier tests for error ($LM_{error}$ $p = 0.646$) and lag ($LM_{lag}$ $p = 0.221$). This indicates that there is no spatial autocorrelation and hence the assumption of independence of residuals of the OLS regression model is not violated. This implies that the OLS model is appropriate for analyzing these county level data and spatial models are not necessary for these data.

## Geographic distribution of significant predictors

The geographic distributions of both the percentage of individuals with asthma (Fig. 5A) and those who are current smokers (Fig. 5B) are similar to that of age-adjusted COPD

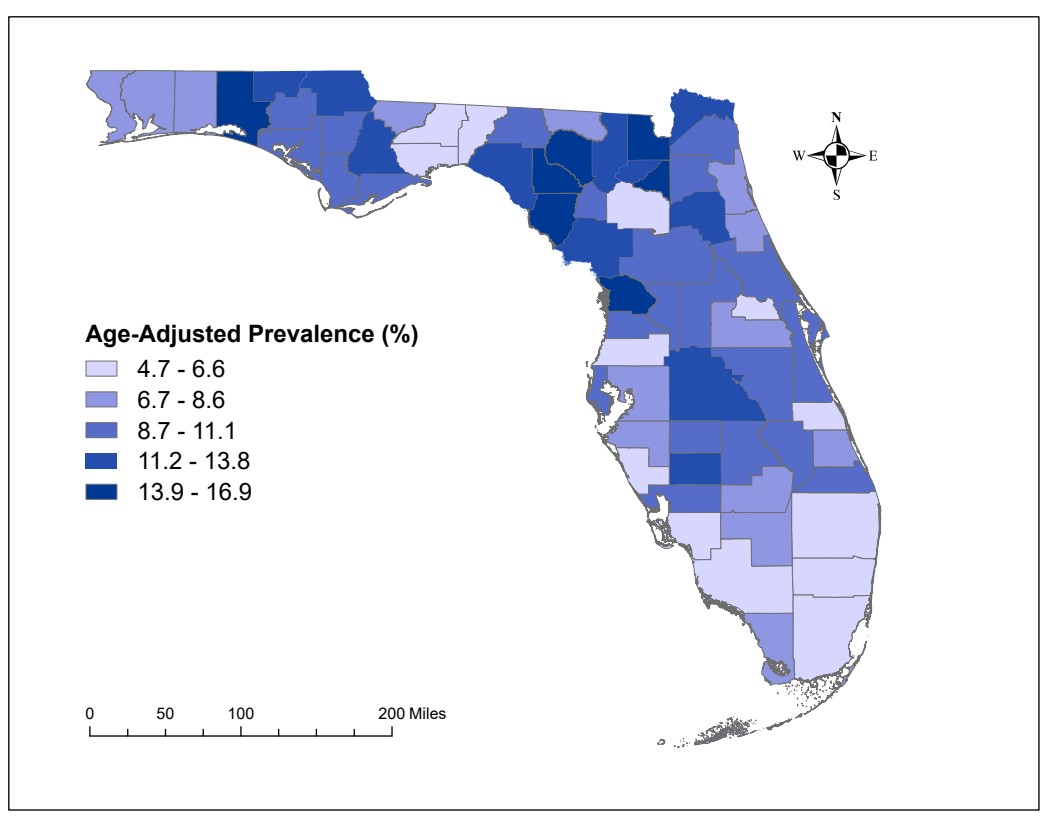

**Figure 3** Geographic distribution of age-adjusted chronic obstructive pulmonary disease prevalence per 100 in Florida, 2019. The base maps were downloaded from the United States Census Bureau: https://www.census.gov/geographies/mapping-files/time-series/geo/tiger-line-file.2019.html; License: Public Domain.

**Table 2** Significant high-prevalence clusters of age-adjusted chronic obstructive pulmonary disease in Florida, 2019.

| Cluster | Observed number of cases | Expected number of cases | Number of counties | Prevalence ratio | *p*-value |
|---|---|---|---|---|---|
| 1 | 334,202 | 243,935 | 13 | 1.37 | 0.001 |
| 2 | 74,174 | 44,165.7 | 10 | 1.68 | 0.001 |
| 3 | 200,116 | 161,968 | 7 | 1.24 | 0.001 |
| 4 | 47,200 | 30,564.2 | 7 | 1.54 | 0.001 |
| 5 | 94,887 | 75,070.9 | 1 | 1.26 | 0.001 |
| 6 | 1,914 | 1,496.96 | 1 | 1.28 | 0.001 |

prevalence (Fig. 3). Counties with the highest percentage of current smokers were primarily located in north-central Florida as well as the panhandle. However, the percentage of current smokers was fairly high across the state, with 29 counties having at least 20% current smokers. Of the counties with the seven highest COPD prevalence (Baker, Bradford, Citrus, Dixie, Lafayette, Suwannee, and Walton), all but Citrus County had more that

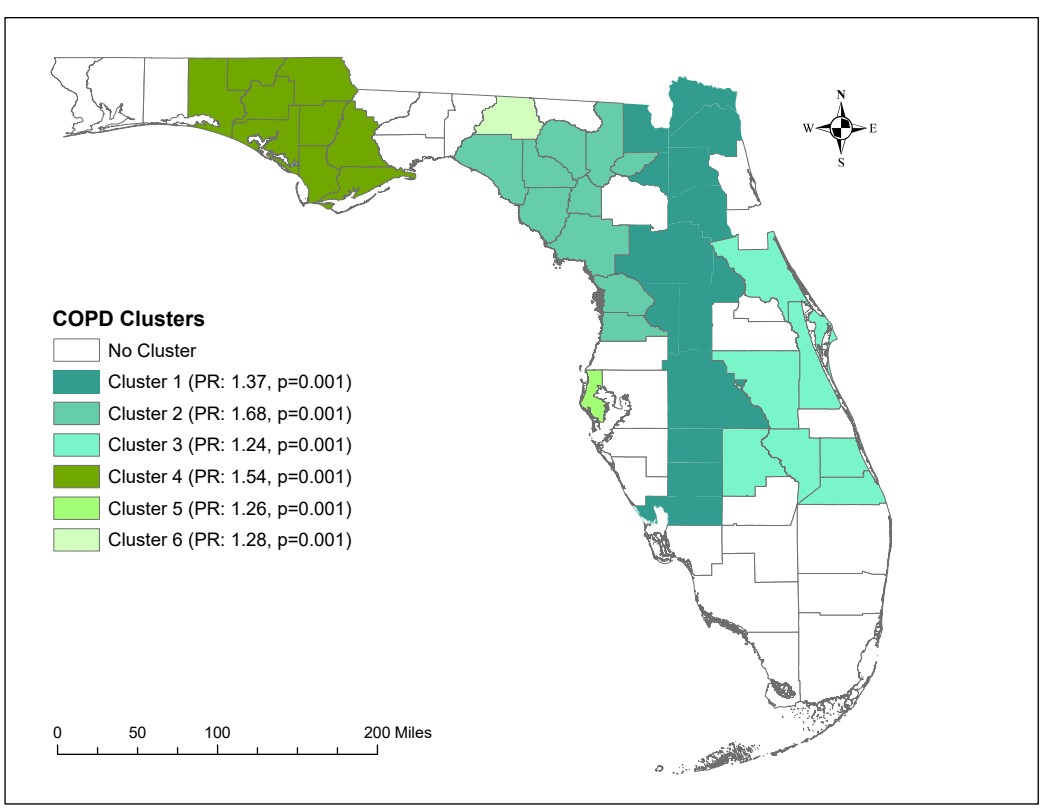

**Figure 4** **Significant clusters of age-adjusted chronic obstructive pulmonary disease Prevalence in Florida, 2019.** The base maps were downloaded from the United States Census Bureau: https://www.census.gov/geographies/mapping-files/time-series/geo/tiger-line-file.2019.html; License: Public Domain.

25% of current smokers. The similarity in the geographic distribution in the prevalence of asthma and that of COPD was less pronounced. The panhandle and north-central Florida generally had high prevalence of both asthma and COPD.

## DISCUSSION

This study investigated geographic clusters and county-level predictors of COPD prevalence in Florida. The results suggest that COPD prevalence varies across geographic areas, which is consistent with the results from other ecological studies of COPD in the US (*Lipton et al., 2005*; *Lipton & Banerjee, 2007*; *Farah, Hosgood & Hock, 2014*; *Croft et al., 2018*; *Blanco et al., 2018*; *Wheaton et al., 2019*). In a state-level study of COPD prevalence, the geographic distribution of COPD was similar to that of current smokers, which is similar to the results of the current analysis (*Wheaton et al., 2019*). A study in Maine reported an overlap between areas of high COPD risk and high asthma risk at the Zip Code Tabulation Area (ZCTA)-level (*Farah, Hosgood & Hock, 2014*). However, that study excluded any non-White cases of COPD from the analysis (*Farah, Hosgood & Hock, 2014*), which may limit the generalizability of the results especially to a more racially or ethnically

**Table 3** Descriptive statistics & univariable associations between age-adjusted chronic obstructive pulmonary disease prevalence and socioeconomic factors in Florida, 2019.

| Variable | Median | 25th quartile | 75th quartile | Unadjusted parameter estimate | 95% confidence interval | p-value |
|---|---|---|---|---|---|---|
| Asthma (%) | 13.3 | 11.3 | 14.8 | 0.455 | 0.213, 0.697 | 0.000372 |
| Current Smokers (%) | 19.3 | 15.7 | 22.3 | 0.465 | 0.373, 0.558 | <0.0001 |
| Former Smokers (%) | 26.8 | 24.4 | 30.1 | 0.002 | −0.142, 0.147 | 0.9750 |
| Never Smoked (%) | 52.2 | 48.3 | 57.2 | −0.251 | −0.340, −0.162 | <0.0001 |
| Mean Daily PM 2.5 | 7.7 | 7.0 | 8.3 | 0.464 | −0.366, 1.295 | 0.2679 |
| Individuals Below the Poverty Line (%) | 14.6 | 12.1 | 20.0 | 0.129 | −0.016, 0.274 | 0.0805 |
| Families Below the Poverty Line (%) | 10.2 | 8.3 | 14.2 | 0.214 | 0.045, 0.382 | 0.0137 |
| Male (%) | 49.4 | 48.7 | 53.2 | 0.356 | 0.163, 0.548 | 0.0005 |
| Female (%) | 50.6 | 46.8 | 51.3 | −0.356 | −0.548, −0.163 | 0.0005 |
| White Hispanic (%) | 8.6 | 5.0 | 18.1 | −0.092 | −0.149, −0.036 | 0.0018 |
| White Non-Hispanic (%) | 71.3 | 59.8 | 77.5 | 0.085 | 0.041, 0.130 | 0.0003 |
| Non-White Hispanic (%) | 1.2 | 0.8 | 1.8 | −1.089 | −1.807, −0.372 | 0.0035 |
| Non-White Non-Hispanic (%) | 16.2 | 11.7 | 22.0 | −0.057 | −0.135, 0.022 | 0.1560 |
| Median Household Income (%) | 53,023.0 | 43,644.0 | 60,264.5 | −0.0001 | −0.00002, −3.121 | 0.0042 |
| Rural Population (%) | 23.8 | 8.5 | 67.5 | 0.023 | 0.0001, 0.046 | 0.0493 |

**Table 4** Significant predictors of age-adjusted chronic obstructive pulmonary disease prevalence in Florida, 2019.

| Variable | Adjusted parameter estimate | 95% confidence interval | p-value |
|---|---|---|---|
| Asthma Diagnosis (%) | 0.200 | 0.004, 0.396 | 0.0457 |
| Current Smoker (%) | 0.408 | 0.305, 0.512 | <0.0001 |
| Non-White Non-Hispanic (%)[a] | −0.047 | −0.101, 0.007 | 0.0858 |

**Notes.**
[a] Included as a confounder for percentage of asthma diagnosis.

diverse population. In a study of state-level prevalence, the geographic distribution of COPD prevalence was similar in current smokers and non-smokers, which the authors hypothesized was due to possible exposure to second-hand smoke among non-smokers (*Wheaton et al., 2019*). Other studies of geographic distributions of COPD have primarily focused on hospitalizations (*Lipton et al., 2005*; *Lipton & Banerjee, 2007*). A California study of COPD hospitalizations showed that the COPD hospitalization rates varied across ZCTAs, particularly across urban and rural areas or areas with varying socioeconomic status (*Lipton et al., 2005*).

In this study, COPD prevalence was associated with the county-level percentage of current smokers and percentage of population with asthma. Although the association between smoking and COPD in individual-level studies is well documented (*Sullivan et al., 2018*; *Wheaton et al., 2019*; *Global Initiative for Chronic Obstructive Lung Disease, 2020*), previous ecological studies are limited in their assessment of area-level risk factors,

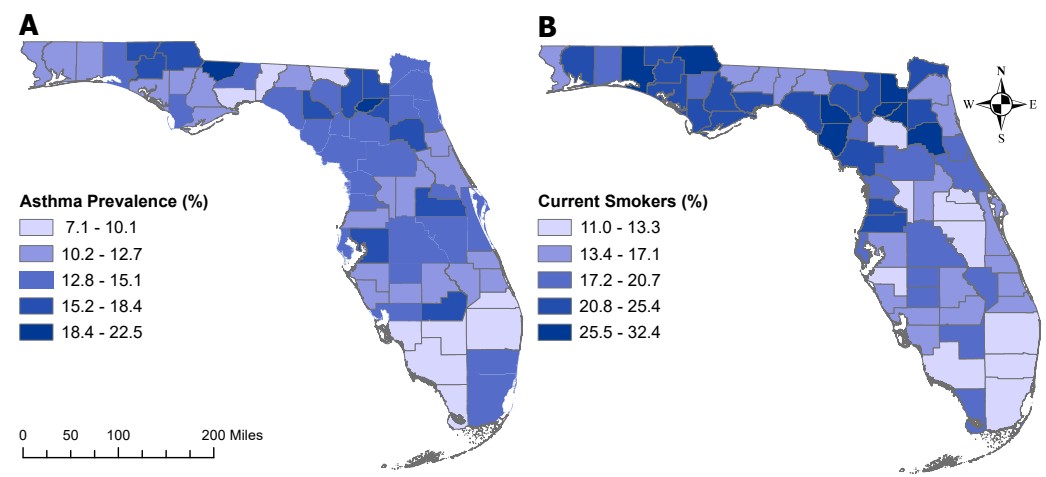

**Figure 5** **Geographic distributions of the percentages of Florida population with asthma (5A) or are current smokers (5B).** The base maps were downloaded from the United States Census Bureau: https://www.census.gov/geographies/mapping-files/time-series/geo/tiger-line-file.2019.html; License: Public Domain.

including smoking status (*Farah, Hosgood & Hock, 2014*; *Kauhl et al., 2018*; *Kumarihamy & Tripathi, 2019*; *Davies, Konings & Lal, 2020*; *Shehu, Farruku & Smaili, 2022*). Previous studies assessing area-level smoking reported no associations with COPD prevalence. The authors hypothesized that the lack of association may be due to methodological limitations (*Farah, Hosgood & Hock, 2014*; *Shehu, Farruku & Smaili, 2022*). At the individual-level, however, smoking is recognized as a primary risk factor for COPD (*Global Initiative for Chronic Obstructive Lung Disease, 2020*), which is similar to the results of the current study although not directly comparable. The results of the current ecological study taken together with the results in the individual-level study suggest that targeting areas with high percentages of current smokers for smoking cessation or prevention campaigns may help reduce the burden of COPD in these areas.

The association between COPD prevalence and asthma is less clear as the mechanism of COPD development among individuals with asthma is not well understood. Only a few individual-level studies have studied the risk of COPD development in those with asthma, both of which found an increase in risk of COPD (*Silva et al., 2004*; *McGeachie et al., 2016*). In ecological studies, asthma prevalence is typically not included in the area-level risk factor investigation (*Kauhl et al., 2018*; *Kumarihamy & Tripathi, 2019*; *Shehu, Farruku & Smaili, 2022*). Only one ecological study reported overlap in high prevalence locations of COPD and asthma, which they suggested could be the result of shared risk factors or co-occurring diseases (*Farah, Hosgood & Hock, 2014*). Although traditionally identified as two distinct diseases, both asthma and COPD are consequences of airway inflammation. A recent review comparing risk factors of COPD and adult-onset asthma identified several risk-factor overlaps between the two conditions. The risk factors common to both conditions included early-life factors (such as low-birth weight and pre-term births), smoking, air pollution, and a number of occupational exposures including dust, pesticides,

and other chemical respiratory irritants (*Holtjer et al., 2023*). In recent years, however, the overlap of asthma and COPD within the same individual has been recognized as a separate syndrome, referred to as the asthma-COPD overlap syndrome (*Leung & Sin, 2017*; *Roman-Rodriguez & Kaplan, 2021*). Thus, it may be possible that the association between county-level percentage of asthma and prevalence of COPD may be the result of the asthma-COPD overlap syndrome. However, the reported prevalence of asthma-COPD overlap syndrome is highly variable across studies primarily due to lack of consensus on the definition of the syndrome, but the reported estimate in the United States general population is approximately 3.7% (*Leung & Sin, 2017*). Regardless, the current results suggest that efforts targeted at reducing asthma prevalence may also be beneficial for reducing COPD prevalence. However, whether these strategies should be focused on shared risk factors or asthma-COPD overlap syndrome is still unclear. Further investigation is needed to elucidate the nature of the association between asthma and COPD in order to better attune prevention efforts.

The percentage of non-White non-Hispanic residents was a confounder for the association between COPD prevalence and percentage of asthma. In the United States, asthma prevalence is higher in the non-White non-Hispanic population compared to the White population (*Pate et al., 2021*). It is suggested that the higher prevalence of asthma among non-White non-Hispanic populations may be due to differences in socioeconomic determinants of health (*Grant, Croce & Matsui, 2022*) some of which are also associated with COPD.

Perhaps, the most surprising result from the study is the lack of association between COPD prevalence and poor air quality and poverty. Previous studies of socioeconomic status have reported significant associations between poverty or lower socioeconomic status and higher COPD prevalence (*Sahni et al., 2017*; *Raju et al., 2019*; *Adeloye et al., 2022*). Unfortunately, the variables used as measures of socioeconomic status vary widely across studies and some of these measures may not completely capture the socioeconomic status within an area, particularly for areas undergoing dynamic changes (*Sahni et al., 2017*). Thus, it is possible that a different measure of socioeconomic status or a smaller unit of analysis, such as census tract might be worth considering for future investigations. Although not directly comparable with the findings of this county-level study, previous individual-level studies have provided some evidence of association between poor air quality and COPD (*Hendryx et al., 2019*; *Shin et al., 2021*). Moreover, poor air quality has been reported to have strong associations with some severe outcomes such as hospitalizations and mortality (*Thurston et al., 2020*). A recent review stated that more evidence is required before poor air quality could be considered a universal risk factor for the onset of COPD (*Thurston et al., 2020*).

## Strengths and limitations

This study used rigorous spatial statistical methods to identify spatial clusters of COPD in Florida. The strength of Tango's Flexible Scan Statistic (FSSS) is its ability to identify irregularly shaped clusters, unlike other methods such as Kulldorff's Spatial Scan Statistics that are better at identifying circular clusters. Additionally, FSSS does not suffer from

multiple comparisons that may be associated with other cluster identification methods such as Moran's Local Indicators of Spatial Associations (LISA) (*Tango & Takahashi, 2005*).

A limitation of this study is that the estimates of COPD prevalence, smoking status, and asthma are all based on self-reporting from health surveys (*Florida Department of Health, 2019*). Self-reported measures may be biased (*Althubaiti, 2016*). Additionally, due to unavailability of data, this study could not investigate the association between COPD prevalence and occupational exposures such as welding fumes or diesel exhaust (*Hart, Eisen & Laden, 2012*; *Koh et al., 2015*). Although the findings of this study are directly valid for Florida, similar study approaches can be used in other geographic locations to help improve our understanding of these disparities and their predictors. Suffice it to say that this study provides important information on the geographic distribution of COPD prevalence and on area-level predictors of COPD prevalence that are imperative for local health planning and resource allocation to address the condition.

## CONCLUSION

There is evidence of spatial clusters of COPD prevalence in Florida. These patterns are explained, in part, by differences in distribution of some health behaviors (smoking) and co-morbidities (asthma). This information is important for guiding intervention efforts to address the condition, reduce health disparities, and improve health for all.

### Funding
The authors received no funding for this work.

### Competing Interests
Agricola Odoi is an Academic Editor for PeerJ.

### Author Contributions
- Sara Howard conceived and designed the experiments, performed the experiments, analyzed the data, prepared figures and/or tables, authored or reviewed drafts of the article, and approved the final draft.
- Agricola Odoi conceived and designed the experiments, authored or reviewed drafts of the article, and approved the final draft.

### Ethics
The following information was supplied relating to ethical approvals (*i.e.*, approving body and any reference numbers):

This study was reviewed by the University of Tennessee, Knoxville Institutional Review Board which determined that it was not human subjects' research (IRB Number: UTK IRB-23-07928-XM). Therefore, it determined that IRB oversight was not required.

## Data Availability

The raw county-level data are available in the Supplementary File.

## Supplemental Information

Supplemental information for this article can be found online at http://dx.doi.org/10.7717/peerj.17771#supplemental-information.

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
