# Peer review of "Spatial patterns and sociodemographic predictors of chronic obstructive pulmonary disease in Florida"

_PeerJ, doi:10.7717/peerj.17771_

## Round 0.1 · original submission · Minor Revisions

Reviewers have recommended further revision.

·

Basic reporting

'no comment

Experimental design

'no comment

Validity of the findings

'no comment

Additional comments

1. Given the recognized differences in smoking rates by sex, and considering smoking's significant role in predicting COPD prevalence, could author explain the rationale why sex is not a predictor in this analysis?
2. Previous studies have established a link between air pollution, socioeconomic factors such as poverty, and COPD prevalence. Given the likely variation in these factors across different counties, it's surprising that this study did not observe similar associations. Could the author clarify the absence of these expected associations in the study's results?
3. Phosphate mining is one of the major industries in Florida, and past research has highlighted the potential COPD risks associated with mining activities. Could the author discuss any potential link between COPD prevalence and regions with significant mining activities, despite this aspect being listed as a limitation?

·

Basic reporting

no comment

Experimental design

no comment

Validity of the findings

no comment

Additional comments

This is an ecological study investigating spatial patterns and predictors of COPD prevalence at the county level. Through multivariable regression, the authors demonstrate that COPD prevalence can be predicted by asthma prevalence and percentage of current smokers. Their findings are consistent with previous literature and our knowledge to date.

Geographic clusters of COPD prevalence are identified. The spatial distribution is similar to that of asthma prevalence and that of percentage of current smokers.

The following are my suggestions:

1. Line 106 to 110, possible reasons for the higher prevalence of COPD in rural areas should be described in this paragraph, such as indoor pollutants.

2. In Table 3, the percentage of female should be close to 50 %. Incorrect data should be corrected.

3. The association between COPD and asthma has biological implications that merit further discussion. In Line 294, please describe the potential shared risk factors (both genetic and environmental) that might contribute to airway inflammation in both COPD and asthma.

Reviewer 3 ·

Basic reporting

The introduction section is well-stated and supported by sufficient evidence; it flows nicely.

Comment 1: I couldn't locate the reference or data source for lines 118-119 and lines 121-122. Therefore, I couldn't verify the reliability of these numbers (Florida Department of Health - Public Health Tracking System).

Experimental design

Line 165: Could you please provide a reference for the statistical testing, specifically the Shapiro-Wilk test?

Line 175: Kindly elaborate on the rationale behind setting a relative risk to 1.2 and discuss the robustness of the study results when varying this value.

Line 188: It is suggested to reconsider decision-making solely based on the p-value, although I believe the analysis was well-conducted. There are many alternative methods to identify predictors.

If possible, could you kindly provide references for the statistics mentioned in lines 193-194 (SW, BP) and 204 (Jenk’s)?

Validity of the findings

Line 191: The identification of the confounder and the interpretation of confounding in line 308 need to be reconsidered. In epidemiology, a confounder should satisfy three criteria, and it has been recommended to identify it with the help of a conceptual framework, rather than solely relying on the model results.

Consideration might be given to listing Table 1 either in the method section or in the appendix.

For Table 3, it is recommended to reformat the unit and place it after the name of the variable. Additionally, present the unadjusted and adjusted risk after exponentiating it for improved readability.

Another limitation to acknowledge is that the results may be applicable primarily to the Florida area, given the absence of an external dataset for validating the study outcomes.

In my opinion, this is a well-structured paper with ample numerical support for decision-making in Florida. The statistical analysis is robust.

---

## Round 0.2 · Major Revisions

Since the analysis is conducted at the county level, using Ordinary Least Squares (OLS) for predictions and Spearman's rank correlation may not be appropriate. You haven't presented your results from spatial autocorrelation. A fundamental assumption of OLS is the independence of observations, but given that your data has a spatial dimension, this assumption might be violated. It is advisable to use a regression method suitable for spatial data.

Please consult a biostatistician to refine your analysis approach. I am open to reevaluating your work once appropriate statistical methods have been applied.

---

## Round 0.3 · accepted · Accept

I confirm that all comments have been adequately addressed.